# Sex specific differences in HIV status disclosure and care engagement among people living with HIV in rural communities in Kenya and Uganda

Chinomnso N. Okorie[1]*, Sarah A. Gutin[2], Monica Getahun[1], Sarah A. Lebu[1], Jaffer Okiring[3], Torsten B. Neilands[2], Sarah Ssali[4], Craig R. Cohen[1], Irene Maeri[5], Patrick Eyul[3], Elizabeth A. Bukusi[1,5], Edwin D. Charlebois[2], Carol S. Camlin[1,2]

1 Department of Obstetrics, Gynecology & Reproductive Sciences, University of California, San Francisco, California, United States of America, 2 Department of Medicine, Division of Prevention Science, University of California, San Francisco, California, United States of America, 3 Infectious Diseases Research Collaboration, Kampala, Uganda, 4 School of Women and Gender Studies, Makerere University, Kampala, Uganda, 5 Kenya Medical Research Institute, Centre for Microbiology Research, Nairobi, Kenya

* Chinomnso.Okorie@ucsf.edu

## Abstract

Non-disclosure of human immunodeficiency virus (HIV) status can hinder optimal health outcomes for people living with HIV (PLHIV). We sought to explore experiences with and correlates of disclosure among PLHIV participating in a study of population mobility. Survey data were collected from 1081 PLHIV from 2015–16 in 12 communities in Kenya and Uganda participating in a test-and-treat trial (SEARCH, NCT#01864603). Pooled and sex-stratified multiple logistic regression models examined associations of disclosure with risk behaviors controlling for covariates and community clustering. At baseline, 91.0% (n = 984) of PLHIV had disclosed their serostatus. Amongst those who had never disclosed, 31% feared abandonment (47.4% men vs. 15.0% women; $p$ = 0.005). Non-disclosure was associated with no condom use in the past 6 months (aOR = 2.44; 95%CI, 1.40–4.25) and with lower odds of receiving care (aOR = 0.8; 95%CI, 0.04–0.17). Unmarried versus married men had higher odds of non-disclosure (aOR = 4.65, 95%CI, 1.32–16.35) and no condom use in the past 6 months (aOR = 4.80, 95%CI, 1.74–13.20), as well as lower odds of receiving HIV care (aOR = 0.15; 95%CI, 0.04–50 0.49). Unmarried versus married women had higher odds of non-disclosure (aOR = 3.14, 95%CI, 1.47–6.73) and lower odds of receiving HIV care if they had never disclosed (aOR = 0.05, 95%CI, 0.02–0.14). Findings highlight gender differences in barriers to HIV disclosure, use of condoms, and engagement in HIV care. Interventions focused on differing disclosure support needs for women and men are needed and may help facilitate better care engagement for men and women and improve condom use in men.

Chinomnso N., Camlin, Carol S., and Gutin, Sarah A. HIV status disclosure and care engagement in rural Kenya and Uganda_2022. Ann Arbor, MI: Inter-university Consortium for Political and Social Research [distributor], 2022-11-12. https://doi.org/10.3886/E182866V1.

**Funding:** This research was supported by the National Institutes of Health, NIMH (R01MH104132 to CS) and the National Institutes of Mental Health of the U.S. Public Health Service (T32 MH19105 to SAG). The study's funders had no role in study design, data collection, data analysis, data interpretation, or writing the report.

**Competing interests:** The authors have declared that no competing interests exist.

## Introduction

Disclosure of one's HIV status has been shown to improve health outcomes among people living with HIV (PLHIV) [1,2] and can reduce human immunodeficiency virus (HIV) transmission by facilitating engagement in HIV-prevention and care [1,3–7]. Alternatively, non-disclosure, which is often a manifestation of HIV-related stigma (whether internalized, anticipated, or in response to enacted stigma), can negatively impact care outcomes [8–10].

HIV disclosure can be both a negative (harmful, stressful, unhelpful) and positive (supportive, and empowering) experience [11]. Positive disclosure experiences are linked to increased social support [12,13], reduced internalized stigma [9], improved mental health [9], and safer sexual behavior practices [14]. Alternatively, it can also be negative, leading to increased stigma, anxiety [2], violence or abuse in relationships [7,11,15–18], abandonment [9,13,19,20], fear of abandonment [13,21], discrimination [2,13,20], and rejection [2,20]. These issues are compounded by limited access to disclosure support, including a lack of feasible disclosure strategies and limited social and peer support, which impedes the benefits of HIV-status disclosure [2,3,13,19,20].

Reasons for PLHIV disclosure or non-disclosure can depend on social relationships, fear of disclosing, and concerns about stigma [11,19]. Disclosure can be instrumental to receiving support (financial, material, moral and emotional, treatment) for those who disclose. Disclosure can be used to explain a change in behavior or appearance and to promote HIV prevention or protect others from HIV. For those who choose not to disclose, it is a way to protect their identity and avoid stigma and discrimination while maintaining a sense of self and safety [8,11]. Some may disclose to avoid involuntary or second-hand disclosure [8]. However, this study focuses on voluntary disclosure (full, selective, or non-disclosure) [8,11]. In addition, PLHIV may disclose to various types of people for various reasons. A study in Uganda found that although a majority of PLHIV reported having disclosed their status, there were significant variations in persons to whom PLHIV disclosed (84% disclosed to family members, 63% to friends, 21% to workplace colleagues, and 18% to others) [8,19].

Further, disclosure is a highly gendered experience, with wide variations in experiences among men and women. Women have often experienced difficulty disclosing their HIV status to intimate partners, fearing negative reactions including violence, blame, and abandonment [2,6,9,19]. While men also fear negative consequences of disclosure including marital conflict, blame, being labeled as promiscuous, and abandonment by their partners, men have generally experienced less severe consequences and benefitted from more social support for disclosure, compared to women [1,18–23]. Further, men have been more likely to disclose to fellow men than women are to other women [3]. Studies in sub-Saharan Africa have also found that men use multiple strategies to avoid disclosure including introducing condom use under the pretext of family planning [2,24–27]. In addition, men often have more freedom compared to women to seek care in remote facilities outside of their community, thereby avoiding disclosure to their partners and families [22].

HIV-status disclosure remains a complex and challenging decision-making process for PLHIV [28–30]. Sex differences in decision-making processes surrounding the choice of individuals to whom to disclose, reasons for non-disclosure, and the effects of these on care engagement are poorly understood, particularly across typologies of relationship types (whether monogamous or concurrent). We sought to explore experiences with and correlates of disclosure among PLHIV participating in a study of population mobility in rural communities in Kenya and Uganda. Findings can be used to document the challenges faced by PLHIV and inform programs designed to reduce stigma and improve health outcomes.

## Materials and methods

### Study design and participants

The *Understanding Mobility and Risk in SEARCH Communities* (R01MH104132) study [31,32] examined mobility, sexual behavior and HIV outcomes in a longitudinal cohort of 2,750 adults in 12 communities participating in a large-scale test-and-treat trial, the Sustainable East Africa Research in Community Health (SEARCH) study (NCT# 01864603), in Kenya and Uganda [29,33]. Methods are described in detail elsewhere [31]; in summary, a stratified random sampling design was used to select the sample of $\sim 200$ individuals from each of 12 SEARCH communities, composed of eight roughly equally-sized groups of sex-specific, HIV-positive and HIV-negative, mobile (away from household six months or more in past 12 months and fewer than half of nights spent in household in past four months) and residentially stable (non-mobile), men and women. HIV-positive individuals and mobile individuals were oversampled to achieve the desired sample size in each stratum. This analysis uses baseline survey data collected from 1081 PLHIV for whom information on HIV status disclosure were available (Fig 1). HIV status disclosure was defined as PLHIV who reported voluntary disclosure of their HIV-positive status to at least one person.

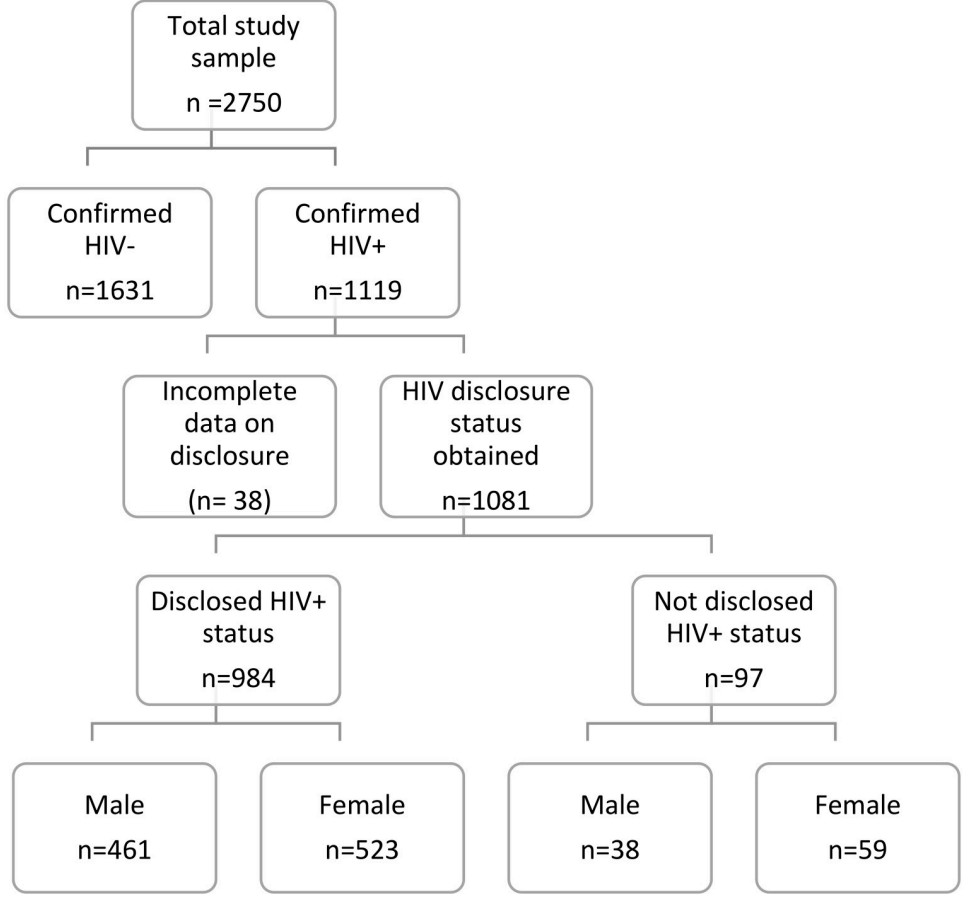

**Fig 1. Analysis flowchart: Study participants' HIV status, and whether disclosed status, at baseline.** A diagram of study participants by HIV status and HIV status disclosure. In a sample of 2750 study paricipants (selected for balance by region, sex, mobility, and HIV status), 1119 were confirmed to be living with HIV, with 91.0% (n = 984) having ever disclosed their HIV status to others (outside a clinic setting) and 9.0% (n = 97) with HIV status not disclosed, at study baseline.

## Procedures

An interviewer-administered survey was used to collect information on household residence (s) and composition, income and livelihoods, histories of migration during childhood and adulthood, patterns of mobility in the past six months (including destinations, reasons, duration and frequency of trips), sexual behavior (including sexual partnership histories over the past five years, using a calendar approach adapted from prior research studies [31,34]), and among PLHIV only, experiences with HIV stigma, disclosure, and engagement in HIV care and treatment. A detailed relationship history calendar permitted measurement of patterns and frequency of condom use, sexual partnership concurrency, and higher-risk partnerships (i.e. any report of a casual partner, commercial sex worker/client, one-night stand, or inherited partner (referring to the Luo practice of widow "inheritance" in which a widow and her children are retained in the family/lineage of her deceased husband; cultural practices include sexual contact with the inheritor [35]), over defined time-periods. Mobility was defined as any overnight travel in the past 6 months.

## Ethics statement

Ethical approvals were received from the University of California San Francisco Committee on Human Research (14–15058), Ethical Review Committee of the Kenya Medical Research Institute (KEMRI/SERU/CMR/3052), Makerere University School of Medicine Research and Ethics Committee (2015–040), and Uganda National Council for Science and Technology (HS 1834). All participants provided written informed consent before taking part in the study.

## Data collection

Data collection was conducted from February 2015-November 2016 by trained research assistants. Surveys were originally developed in English and then translated and administered (using portable tablets) in the local languages (Lusoga, Lugwere, Ateso, Runyankole, and Dholuo) of the research participants, following procedures for the protection of privacy and confidentiality. Surveys took 30 to 90 minutes to complete, were administered in a private area, and participants and interviewers were gender-matched to maximize comfort with sharing sensitive information. Participants were compensated the equivalent of $5 USD for their time and/or transport reimbursement, which is standard procedure in the region.

## Data analyses

Of the confirmed PLHIV at baseline (n = 1119), complete data for this analysis were available for 1081 participants (499 men, 582 women) (Fig 1). Descriptive statistics were computed to explore HIV status disclosure patterns by sex. Bivariate comparisons that accounted for clustering of individuals within communities (Rao-Scott F-tests) were used to characterize the relationship between HIV-positive status disclosure and sex. Pooled, and sex-stratified mixed effects logistic regression models were fitted to examine associations of HIV-positive status non-disclosure with sexual risk behaviors including past 6-month condom use, past year relationship concurrency, and HIV care engagement controlling for age, marital status, region and adjusted for clustering at the community-level. These factors were included in models because prior research has suggested an association between disclosure and age and marital status [36–38]. The sample in multivariable models was smaller (n = 906) because of case-wise deletion for missing data. All analyses were conducted using Stata statistical software version 16.1 (College Station, TX, USA).

## Results

### Characteristics of the sample

Of 1081 PLHIV included in the baseline analyses, 91.0% (n = 984; 523 female, 461 male) reported disclosing their status to at least one person and 9.0% (n = 97; n = 59 female, n = 38 male) reported non-disclosure of their HIV status to anyone (other than their healthcare provider) (Table 1). Overall, the majority of respondents (73.1%) were married, had received some primary-level education (83.2%), and were involved in low HIV-risk occupations (79.1%).

### PLHIV who have disclosed their HIV status

The majority of those who reported ever disclosing their status were female (53.2%, n = 523) and 74.5% were currently married (Table 1). The mean age of those who had disclosed was 42 years. Further, 60.2% reported any past 6-month condom use and 22.0% reported being in concurrent relationships in 2015–16. Almost all of those who had disclosed (97.7%) were receiving HIV care, were enrolled in antiretroviral therapy (ART) programs (95.9%), were taking ART (94.5%), and were attending a clinic (97.6%). In addition, 38.1% (n = 375) of mobile PLHIV disclosed while 61.9% (n = 609) of non-mobile PLHIV had disclosed (Table 1).

Among PLHIV who had disclosed, the first person to whom they had most commonly disclosed was a spouse/partner (59.4%) (Fig 2). When stratified by sex, differences were seen in whom men and women disclosed to. Overall, men were significantly more likely to disclose to a spouse/partner first, compared to women (78.7% vs. 42.0%, $F_{(4.18, 45.94)}$ = 20.7750, $p<0.001$) (Fig 2). However, even among those who reported disclosing their status to at least one person, 21.4% wished to disclose to others but felt they could not (Table 2). The majority of women wanted to disclose to a friend (25%), and/or mother (25%), while men preferred to disclose to a brother (33%), or friend (31%) (Table 2). The main reasons for not disclosing to additional people were fear of being judged (56.9%), fear of abandonment (27.0%), and other reasons (25.6%) (Table 2). The only significant differences observed between men and women were feeling guilty for extramarital affairs (15.5% men vs. 3.5% women, $F_{(1, 11)}$ = 5.985, $p$ = 0.032), or other reasons including not trusting others and fears of hurting/stressing others (9.3% men vs. 39.5% women, $F_{(1, 11)}$ = 18.717, $p$ = 0.001) (Table 2).

### PLHIV who have not disclosed their HIV status

Of the 9.0% (n = 97) of PLHIV who had not disclosed their status, the majority (60.8%) were female and reported being currently married (39.2%, n = 38) (Table 1). Further bivariate analyses among PLHIV who had not disclosed their status showed that 37.7% (n = 29) reported any past 6-month condom use and 3.2% reported being in concurrent relationships. HIV care engagement was lower among those who had not disclosed compared to those who had disclosed, with 78.4% receiving HIV care, 73.2% enrolled in ART programs and taking ART (n = 71 for each), and 97.3% attending one clinic (Table 1). In addition, of the participants who had not disclosed their HIV status, 44.3% were mobile compared to 55.7% who were non-mobile (Table 1). The main reasons for not disclosing HIV status included being afraid of being judged (54.6%) and afraid of being abandoned (30.9%) (Table 3). Significant differences were observed between men and women with men being more afraid of being abandoned than women (47.4% men vs. 15% women, $F_{(1, 11)}$ = 12.440, $p$ = 0.005), and more men (27.5%) than women (15.4%) having other reasons (e.g. not being interested in disclosing (30.8%) ($F_{(1, 11)}$ = 13.452, $p$ = 0.004) (Table 3).

**Table 1. Characteristics of adults who had and had not disclosed HIV-positive status at baseline, adjusted for community clustering.**

| Characteristics | Overall (n = 1081) | | Non- Disclosed (n = 97) | | Disclosed (n = 984) | | F (df) | p |
|---|---|---|---|---|---|---|---|---|
| | n | % | n | % | n | % | | |
| **Demographics** | | | | | | | | |
| *Age (Mean, SE)* | 42.2 | 0.47 | 40.1 | 1.06 | 42.4 | 0.55 | F(1, 11) = 3.00 | 0.111 |
| *Sex* | | | | | | | F(1, 11) = 1.837 | 0.202 |
| Male | 499 | 46.2 | 38 | 39.2 | 461 | 46.9 | | |
| Female | 582 | 53.8 | 59 | 60.8 | 523 | 53.2 | | |
| *Occupational-Risk categories* | | | | | | | F(1.69,18.62) = 0.959 | 0.388 |
| Low risk | 855 | 79.1 | 72 | 78.3 | 783 | 82.6 | | |
| High risk | 184 | 17.0 | 19 | 20.7 | 165 | 17.4 | | |
| *Educational Attainment* | | | | | | | F(1.95, 21.41) = 0.575 | 0.567 |
| No School/missing | 144 | 13.7 | 11 | 11.6 | 133 | 13.9 | | |
| Some primary/up to completed | 874 | 83.2 | 82 | 86.3 | 792 | 82.9 | | |
| Some secondary or beyond | 33 | 3.1 | 2 | 2.1 | 31 | 3.2 | | |
| *Marital Status* | | | | | | | F(1, 11) = 10.481 | **0.008** |
| Currently married | 789 | 73.1 | 57 | 58.8 | 732 | 74.5 | | |
| Other * | 290 | 26.9 | 40 | 41.2 | 250 | 25.5 | | |
| **Migration history** | | | | | | | | |
| *Any migration in past 5 years* | | | | | | | F(1, 11) = 1.228 | 0.292 |
| No | 412 | 38.1 | 41 | 42.3 | 371 | 37.7 | | |
| Yes | 669 | 61.9 | 56 | 57.7 | 613 | 62.3 | | |
| *Any past 1 year migration* | | | | | | | F(1, 11) = 1.172 | 0.302 |
| No | 1002 | 92.7 | 93 | 95.9 | 909 | 92.4 | | |
| Yes | 79 | 7.3 | 4 | 4.1 | 75 | 7.6 | | |
| *Any past 2 year migration* | | | | | | | F(1, 11) = 0.691 | 0.424 |
| No | 943 | 87.2 | 88 | 90.7 | 855 | 86.9 | | |
| Yes | 138 | 12.8 | 9 | 9.3 | 129 | 13.1 | | |
| **Mobility patterns** | | | | | | | | |
| *Any past 6 month work travel* | | | | | | | F(1, 11) = 0.913 | 0.360 |
| No | 937 | 86.7 | 87 | 89.7 | 850 | 86.4 | | |
| Yes | 144 | 13.3 | 10 | 10.3 | 134 | 13.6 | | |
| *Any 6 past month non-work travel* | | | | | | | F(1, 11) = 2.337 | 0.155 |
| No | 595 | 55.0 | 60 | 61.9 | 535 | 54.4 | | |
| Yes | 486 | 45.0 | 37 | 38.1 | 449 | 45.6 | | |
| *Any past 6 month (overnight) travel* | | | | | | | F(1, 11) = 1.365 | 0.267 |
| Mobile | 418 | 38.7 | 43 | 44.3 | 375 | 38.1 | | |
| Non-mobile | 663 | 61.3 | 54 | 55.7 | 609 | 61.9 | | |
| **Sexual behavior** | | | | | | | | |
| *Any condom use in past 6 months* | | | | | | | F(1, 11) = 8.605 | **0.014** |
| Yes condom use | 529 | 58.3 | 29 | 37.7 | 500 | 60.2 | | |
| Never | 379 | 41.7 | 48 | 62.3 | 331 | 39.8 | | |
| *Any high-risk partnerships, 2015–16** | | | | | | | F(1, 11) = 0.5626 | 0.469 |
| No | 923 | 85.4 | 81 | 83.5 | 842 | 85.6 | | |
| Yes | 158 | 14.6 | 16 | 16.5 | 142 | 14.4 | | |
| *Any concurrent partnerships, past 6 months*** | | | | | | | F(1, 11) = 9.2789 | **0.011** |
| No | 922 | 79.2 | 92 | 9.9 | 830 | 78.0 | | |
| Yes | 159 | 20.8 | 5 | 3.2 | 154 | 22.0 | | |

(*Continued*)

**Table 1.** (Continued)

| Characteristics | Overall (n = 1081) | | Non- Disclosed (n = 97) | | Disclosed (n = 984) | | F (df) | p |
|---|---|---|---|---|---|---|---|---|
| | n | % | n | % | n | % | | |
| **HIV care engagement** | | | | | | | | |
| *Currently receiving HIV Care* | | | | | | | F(1, 11) = 42.580 | **<0.001** |
| No | 46 | 4.3 | 21 | 21.7 | 25 | 2.5 | | |
| Yes | 1035 | 95.7 | 76 | 78.4 | 959 | 97.5 | | |
| *Ever enrolled in ART program* | | | | | | | F(1.55,17.01) = 31.509 | **<0.001** |
| No | 66 | 6.1 | 26 | 26.8 | 40 | 4.1 | | |
| Yes | 1015 | 93.9 | 71 | 73.2 | 944 | 95.9 | | |
| *Currently taking ART* | | | | | | | F(1.60,17.55) = 23.927 | **<0.001** |
| No | 80 | 7.4 | 26 | 26.8 | 54 | 5.5 | | |
| Yes | 1001 | 92.6 | 71 | 73.2 | 930 | 94.5 | | |
| *Attending clinic* | | | | | | | F(1, 11) = 0.0247 | 0.878 |
| One clinic | 986 | 97.5 | 71 | 97.3 | 915 | 97.6 | | |
| More than one clinic | 25 | 2.5 | 2 | 2.7 | 23 | 2.5 | | |
| *Ever missed appointments/ Dropped out of care for a time* | | | | | | | F(1.81, 19.91) = 26.754 | **<0.001** |
| No | 928 | 89.7 | 69 | 90.8 | 859 | 89.6 | | |
| One or more appointments | 107 | 10.3 | 7 | 1.0 | 100 | 10.4 | | |

Bivariate comparisons that accounted for clustering of individuals within communities (Rao-Scott F-tests) were used to examine the relationship between selected characteristics and disclosure of HIV status, at baseline. Data are column percentages; percentages sum to greater than 100% because multiple responses were permitted.

*Marital status, "Other": Single, widowed, divorced, separated, do not know, missing.

** Higher-risk partnerships: Any casual partner, commercial sex worker/client, one-night stand, and inherited partner.

*** Concurrent partnerships: Any overlapping sexual partners within any month in the period (2015–16).

## Analysis of HIV non-disclosure patterns

Multivariable analyses using sex-pooled and sex stratified logistic regression models revealed that compared to those who were married, those who were unmarried had three times the odds of not disclosing an HIV-positive status (aOR = 3.17; 95%CI, 1.69–5.94, p<0.001). PLHIV reporting no past 6-month condom use had over two times the odds of non-disclosure relative to those reporting any condom use during the period (aOR = 2.44; 95%CI, 1.40–4.25, p = 0.002). Those reporting concurrent partnerships also had lower odds of non-disclosure compared to those in monogamous relationships (aOR = 0.37; 95%CI, 0.14–0.99, p = 0.047). Furthermore, those currently receiving HIV care had lower odds of non-disclosure relative to those not receiving care (aOR = 0.08; 95%CI, 0.04–0.17, p<0.001).

In sex-stratified models, men who were unmarried compared to married had almost five times the odds of not disclosing an HIV-positive status (aOR = 4.65; 95%CI, 1.32–16.35, p = 0.017). Also, men with no past 6-month condom use had almost five times the odds of non-disclosure compared to those who reported any condom use (aOR = 4.80; 95%CI, 1.74–13.20, p = 0.002) and men receiving HIV care compared to men not in care had lower odds of non-disclosure (aOR = 0.15; 95% CI, 0.04–0.49, p = 0.002). Women who were unmarried compared to married had three times the odds of not disclosing an HIV-positive status (aOR = 3.14; 95%CI, 1.47–6.73, p = 0.003). In addition, women receiving HIV care compared to those not in care had 95% lower odds of non-disclosure (aOR = 0.05; 95%CI, 0.02–0.14, p<0.001) (Table 4).

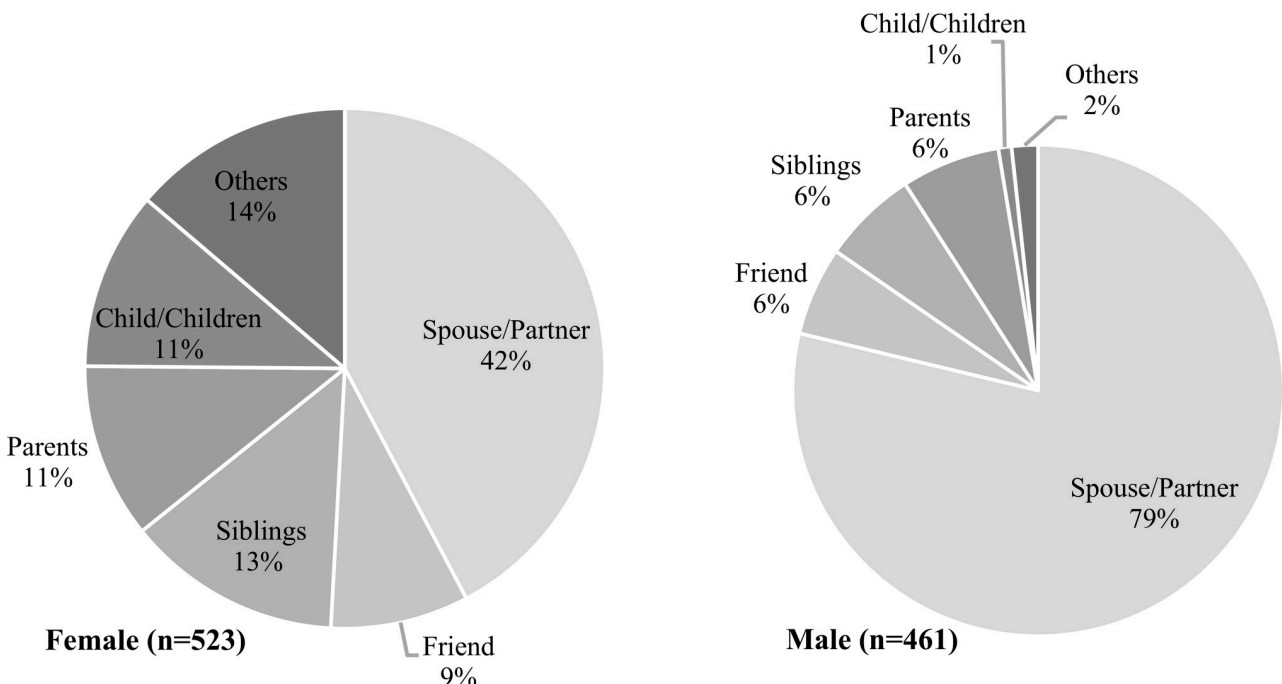

**Fig 2. Types of individuals to whom PLHIV first disclosed their HIV status, by sex (n = 984).** The graph shows the proportion (%) of individuals to whom PLHIV first disclosed their HIV status (n = 984, $F_{(4.18, 45.94)}$ = 20.7750, p<0.001). Data shown are coloumn percentages, by sex. Test statistic is adjusted for clustering at the community-level.

## Discussion

This study highlights the impact of sex-specific patterns of HIV-disclosure amongst a cohort of PLHIV in the context of rapid ART expansion in rural Kenya and Uganda. At baseline, the majority of PLHIV had disclosed their status to at least one person, predominently a spouse/partner. Even among those who had disclosed, both men and women expressed the desire to further disclose their status to a close friend or family member. Those who had never disclosed had higher odds of being unmarried and never using condoms and lower odds of HIV care engagement. When stratified by sex, we found that men who have never disclosed have higher odds of being unmarried and reporting no condom-use in the past 6-months and lower odds of HIV care engagement, while women who had never disclosed their HIV status had higher odds of being unmarried and lower odds of HIV care engagement.

With the aid of community researchers and meticulous community engagement in rural Kenya and Uganda, this study was able to attain an HIV-status disclosure rate over 90% at baseline. This finding aligns with other studies conducted in Cameroon, Nigeria, Malawi, and Zimbabwe that have reported disclosure rates above 80% [29,39–44]. Even with high disclosure rates in this study, disclosing was not without its emotional, relational, and psychological hardships [45]. Among PLHIV who had already disclosed their status, we found that many wished to disclose to others, but feared judgment and/or felt guilty for extramarital affairs. Research has established that there are negative consequences of disclosing, including stigma, anxiety, fear of abandonment, discrimination, rejection, and unhealthy relationships with others [2,3,13,19,20]. Therefore, a lack of access to suitable disclosure strategies and a lack of peer/social support can continue to limit HIV status disclosure and an individual's well-being.

**Table 2. Patterns of HIV disclosure among PLHIV who have already disclosed HIV status, by sex and adjusted for community clustering.**

| | Overall | % | Female | % | Male | % | F (df) | p |
|---|---|---|---|---|---|---|---|---|
| | (n = 984) | | (n = 523) | | (n = 461) | | | |
| **PLHIV wished to disclose to others to whom they had not yet disclosed** | | | | | | | | |
| No | 773 | 78.6 | 409 | 78.2 | 364 | 79.0 | F(1, 11) = 0.024 | 0.879 |
| Yes | 211 | 21.4 | 114 | 21.8 | 97 | 21.0 | | |
| | Overall | % | Female | % | Male | % | | p |
| | (n = 211) | | (n = 114) | | (n = 97) | | | |
| **Other people to whom PLHIV would like to disclose HIV status, but felt they could not*** | | | | | | | | |
| Friend | 59 | 28.0 | 29 | 25.4 | 30 | 30.9 | F(1, 11) = 0.591 | 0.458 |
| Mother | 54 | 25.6 | 28 | 24.6 | 26 | 26.8 | F(1, 11) = 0.357 | 0.563 |
| Brother | 44 | 20.9 | 12 | 10.5 | 32 | 33.0 | F(1, 11) = 7.377 | **0.020** |
| Sister | 41 | 19.4 | 16 | 14.0 | 25 | 25.8 | F(1, 11) = 2.615 | 0.134 |
| Spouse/partner | 35 | 16.6 | 23 | 20.2 | 12 | 12.4 | F(1, 11) = 9.265 | **0.011** |
| Other relatives | 25 | 11.8 | 20 | 17.5 | 5 | 5.2 | F(1, 11) = 4.241 | 0.064 |
| Child/children | 19 | 9.0 | 11 | 9.6 | 8 | 8.2 | F(1, 11) = 0.117 | 0.739 |
| Father | 15 | 7.1 | 9 | 7.9 | 6 | 6.2 | F(1, 11) = 0.672 | 0.430 |
| Others | 8 | 3.8 | 7 | 6.1 | 1 | 1.0 | F(1, 11) = 3.835 | 0.076 |
| Employer | 4 | 1.9 | 0 | 0.0 | 4 | 4.1 | F(1, 11) = 2.741 | 0.126 |
| **Reasons for not disclosing further*** | | | | | | | | |
| Afraid of being judged | 120 | 56.9 | 49 | 43.0 | 71 | 73.2 | F(1, 11) = 4.255 | 0.064 |
| Afraid of being abandoned | 57 | 27.0 | 27 | 23.7 | 30 | 30.9 | F(1, 11) = 0.469 | 0.508 |
| Afraid of violence | 25 | 11.8 | 16 | 14.0 | 9 | 9.3 | F(1, 11) = 0.938 | 0.354 |
| Felt guilty (extramarital affair) | 19 | 9.0 | 4 | 3.5 | 15 | 15.5 | F(1, 11) = 5.985 | **0.032** |
| Not disclosed for other reasons** | 54 | 25.6 | 45 | 39.5 | 9 | 9.3 | F(1, 11) = 18.717 | **0.001** |

Bivariate comparisons that accounted for clustering of individuals within communities (Rao-Scott F-tests) were used to examine associations of disclosure characteristics by sex; data are column percentages.

*Percentages sum to greater than 100% because multiple responses were permitted.

**Other reasons includes being distrustful of others, fear of hurting/stressing others, not found the right time to disclose, not interested in disclosing, those advised by health worker/others not to disclose, afraid of being ridiculed/made fun of, afraid of losing job, current partner discourages disclosure, and those not ready to disclose.

In this study, the majority of PLHIV had disclosed to at least one person, most commonly a spouse/partner. In addition, we saw some differences by sex, with men overwhelmingly disclosing to their spouse/partner (79% compared to 42% for women). It is possible that men disclose to partners/spouses more than women because they feel greater social support from their female parters. Social support is an important aspect of psychological adjustment that can promote well-being for many PLHIV [46,47] and is an essential resource for coping [46]. However, social support varies by sex. Men who disclose tend to benefit from increased social support [22] and the consequences of disclosure are less severe [2,23]. However, amongst young women, disclosure can be more difficult as many fear negative reactions including upsetting a partner, violence, abandonment, and blame [2,19,41]. Therefore, partner notification policies and support programs must be responsive to the potential negative consequences associated with disclosure for women [48]. This highlights the need, particularly for women, for facilitated couples disclosure. In addition, even though men benefit more than women from increased social support following disclosure of their HIV-status [22], there is a need for male-centered interventions in HIV care [42], because men have reported increased care and

**Table 3. Reasons for non-disclosure of HIV status, by sex and adjusted for community clustering.**

| Reasons for non-disclosure* | Overall (n = 97) | % | Female (n = 59) | % | Male (n = 38) | % | F (df) | p |
|---|---|---|---|---|---|---|---|---|
| Afraid of being judged | 53 | 54.6 | 29 | 36.3 | 24 | 63.2 | F(1, 11) = 0.569 | 0.467 |
| Afraid of being abandoned | 30 | 30.9 | 12 | 15.0 | 18 | 47.4 | F(1, 11) = 12.440 | **0.005** |
| Afraid of violence | 17 | 17.5 | 13 | 16.3 | 4 | 10.5 | F(1, 11) = 1.196 | 0.297 |
| Felt guilty | 8 | 8.2 | 4 | 5.0 | 4 | 10.5 | F(1, 11) = 0.504 | 0.493 |
| Afraid for other reasons | 26 | 26.8 | 22 | 27.5 | 4 | 15.4 | F(1, 11) = 13.452 | **0.004** |
| **Other Reasons for non-disclosing (specified)** | Overall (n = 97) | % | Female (n = 59) | % | Male (n = 38) | % | F (df) | p |
| Denial/Non Acceptance | 6 | 23.1 | 6 | 27.3 | 0 | 0.0 | F(2.41, 19.30) = 0.630 | 0.572 |
| Fear Of Hurting/Stress | 4 | 15.4 | 3 | 13.6 | 1 | 25.0 | | |
| Not Interested In Disclosure | 8 | 30.8 | 7 | 31.8 | 1 | 25.0 | | |
| Other** | 8 | 30.8 | 6 | 27.3 | 2 | 50.0 | | |

Bivariate comparisons that accounted for clustering of individuals within communities (Rao-Scott F-tests) were used to examine reasons for not disclosing HIV status by sex; data are row percentages. Question stem for PLHIV who have not disclosed (n = 97): "Can you tell the reason why you haven't felt able to disclose your status to anyone?", 108 PLHIV responded to one or more reasons for not disclosing status (58 females and 50 males).

*percentages sum to greater than 100% because multiple responses were permitted.

**Other includes Distrusting of Others, No one to disclose to, Afraid of losing job.

**Table 4. Multivariate analysis examining factors associated with non-disclosure at basline, straified by sex and adjusted for community clustering.**

| | TOTAL n = 906 | | | | MEN n = 471 | | | | WOMEN n = 435 | | | |
|---|---|---|---|---|---|---|---|---|---|---|---|---|
| | Unadjusted Models * | | | | Adjusted Models | | | | Adjusted Models | | | |
| | OR | 95% CI | | p | aOR | 95% CI | | p | aOR | 95% CI | | p |
| **Age (continuous)** | 0.98 | 0.96 | 1.00 | 0.107 | 1.00 | 0.97 | 1.04 | 0.890 | 0.96 | 0.92 | 0.99 | **0.017** |
| **Region** | | | | | | | | | | | | |
| Kenya | REF | - | - | - | - | - | - | - | - | - | - | - |
| Uganda E | 1.82 | 0.89 | 3.70 | 0.099 | 2.09 | 0.72 | 6.02 | 0.173 | 1.46 | 0.52 | 4.13 | 0.472 |
| Uganda SW | 1.71 | 0.92 | 3.18 | 0.088 | 1.78 | 0.63 | 5.02 | 0.274 | 1.60 | 0.72 | 3.52 | 0.248 |
| **Sex** | | | | | | | | | | | | |
| Male | REF | - | - | - | - | - | - | - | - | - | - | - |
| Female | 1.25 | 0.71 | 2.20 | 0.441 | - | - | - | - | - | - | - | - |
| **Marital status** | | | | | | | | | | | | |
| Currently married | REF | - | - | - | REF | - | - | - | REF | - | - | - |
| *Other | 3.17 | 1.69 | 5.94 | **<0.001** | 4.65 | 1.32 | 16.35 | **0.017** | 3.14 | 1.47 | 6.73 | **0.003** |
| **Any condom use in last 6 months** | | | | | | | | | | | | |
| Yes condom use | REF | - | - | - | REF | - | - | - | REF | - | - | - |
| Never | 2.44 | 1.40 | 4.25 | **0.002** | 4.80 | 1.74 | 13.20 | **0.002** | 1.81 | 0.89 | 3.70 | 0.104 |
| **Any Concurrent Partnership 2015–16** | | | | | | | | | | | | |
| No | REF | - | - | - | REF | - | - | - | REF | - | - | - |
| Yes | 0.37 | 0.14 | 0.99 | **0.047** | 0.53 | 0.17 | 1.65 | 0.276 | 0.14 | 0.02 | 1.22 | 0.075 |
| **Receiving HIV care** | | | | | | | | | | | | |
| No | REF | - | - | - | REF | - | - | - | REF | - | - | - |
| Yes | 0.08 | 0.04 | 0.17 | **<0.001** | 0.15 | 0.04 | 0.49 | **0.002** | 0.05 | 0.02 | 0.14 | **<0.001** |

Multivariate Analysis using mixed effect logistic regression to measure non-disclosure, stratified by sex at baseline and adjusted for community clustering.

*Marital status-other = (single, widowed, divorced, separated, do not know, missing).

support when they disclose to fellow men compared to when women disclose to women [3]. Another possible next step is to create interventions and treatments that support the family unit (mother, father, and child) as a focal point to increase HIV disclosure and care engagement and to motivate more HIV disclosure to spouses, particularly among women, either before or after initiating ART.

The risks of disclosure are particularly salient for women. More women living with HIV in this study who had not disclosed their status reported fears of violence and judgement as their primary concern for non-disclosure whereas men reported being afraid of abandonment and judgment. This coincides with qualitative research from Kenya and Uganda which has found that negative consequences of disclosure, including severe consequences (i.e. violence), were reported disproportionately by women [2]. In that study, women expressed anxieties around partner abandonment or violence and perceived greater HIV/AIDS stigma [39], whereas men were concerned about their partners perceiving them as promiscuous [2,14,16,39]. Disclosure approaches within relationships are needed as successful disclosure within intimate partnerships can lead to engagement in risk-reduction strategies [2].

Sex-specific factors associated with HIV disclosure patterns need interventions to go beyond a majority of PLHIV reporting disclosure to a spouse/partner. This includes creating a safe space for both sexes to equally express their status and further disclose to others within their family/extended family. Expanding support for assisted disclosure for couples and families is critical and efforts to strengthen health systems capacity for clinician or counselor-assisted disclosure is needed. Yet such strategies should be gender-sensitive and attuned to men's and women's differing needs and experiences. For example, interventions to encourage and support women in safely disclosing their status can focus on restructuring comprehensive support services and re-training peer educators–a crucial support element in creating a safe disclosure environment for women.

Furthermore, factors associated with non-disclosure vary by sex and include marital status, condom use, and engagement with HIV care. Unmarried men (single, divorced or widowed) had five times the odds of non-disclosure compared to those who were married. Those who are married do not have to contend with the same level of fear as those who are unmarried. In addition, men who had not disclosed were not using condoms. This effect was not seen among women. It is possible that men feared that using condoms might signal their HIV-positive status to a partner or lead to stigma, as prior literature among men who have sex with men has suggested [49]. Research suggests that sometimes men have used avoidant disclosure strategies (i.e. introduction of condom use) under the pretext of family planning and protecting their partners while women have used such techniques when they were unsure of their partner's HIV status [2]. it is usually easier for men living with HIV than women to disclose based on gender differences in sexual decision-making power [2]. Women have reported substantial difficulties in negotiating condom use and may not culturally regard using a condom as a sense of empowerment and control over their own bodies [15]. This finding suggests the importance of disclosure to ensure optimal risk-reduction techniques such as condom use and HIV care engagement.

In addition, among both women and men, the odds of non-disclosure were reduced among those receiving HIV care. HIV status non-disclosure can play a critical role in care disruption resulting in the inability or reluctance to take medications or attend clinic, for fear of disclosing one's status [8,19]. The odds of non-disclosure were lower for men than for women who were engaged in care. This can be linked to existing evidence that found that men were often enrolled in HIV care secretively or at distant clinics while women found challenges initiating or staying engaged in care [2].

Prior research has highlighted that mobility is a highly gendered experience that is associated with higher-risk sexual behaviors [50]. Migrants who are more mobile engage in higher-

risk sexual behaviors while travelling, increasing their risk of post-migration HIV-acquisition [51]. In this study, neither mobility or migration affected HIV disclosure rates at baseline. Nevertheless, HIV disclosure responses were different for men and women.

## Limitations

The cross-sectional nature of the study limits our ability to draw causal inferences. All community participants in this study were PLHIV attending clinical care in communities undergoing rapid ART scale up as a part of a community-based intervention study, thus limiting generalizability; however, study contexts are illustrative of similar communities in high HIV prevalence regions with varied rates of disclosure, mobility, access to care, and sex-specific disclosure issues.

## Conclusion

This study highlights the substantial gender differences and barriers to HIV disclosure. The findings demonstrate a need for attention to the differing disclosure experiences and support needs for both women and men in East Africa despite mobility. As the experience of this test-and-treat trial demonstrates, the rapid scale-up of HIV testing and ART rollout makes a critical mass of individuals newly aware of their HIV diagnosis, and newly presented with the dilemmas of disclosure, rendering the need for a robust programmatic response all the more urgent. However, increasing the number of people who have disclosed is only possible in a conducive environment. Governments and AIDS organizations must refocus and reform programs/services to provide adequate emotional and optimal organizational support to those who disclose, including peer support, counseling, and providing adequate training for health workers to offer proper HIV care and counseling [5]. Creating a safe space for disclosure, particularly for women disclosing to partners, is an area of priority within the context of rapid ART expansion.

## Supporting information

**S1 Text. Inclusivity questionnaire: PLOS questionnaire on inclusivity in global research.** (DOCX)

## Acknowledgments

We wish to acknowledge the SEARCH Principal Investigators, Co-Investigators, and members of SEARCH study teams in all regions for their contributions to this research. We thank the Ministries of Health of Kenya and Uganda for their ongoing partnership in the research. Finally, we express our gratitude to study participants for their contribution to this research.

## Author Contributions

**Conceptualization:** Monica Getahun, Torsten B. Neilands, Elizabeth A. Bukusi, Edwin D. Charlebois, Carol S. Camlin.

**Data curation:** Monica Getahun, Jaffer Okiring, Irene Maeri, Patrick Eyul.

**Formal analysis:** Chinomnso N. Okorie, Sarah A. Gutin, Jaffer Okiring, Edwin D. Charlebois.

**Investigation:** Monica Getahun, Sarah Ssali.

**Project administration:** Monica Getahun.

**Supervision:** Monica Getahun, Sarah Ssali, Carol S. Camlin.

**Validation:** Chinomnso N. Okorie.

**Visualization:** Chinomnso N. Okorie.

**Writing – original draft:** Chinomnso N. Okorie, Sarah A. Gutin, Monica Getahun.

**Writing – review & editing:** Chinomnso N. Okorie, Sarah A. Gutin, Monica Getahun, Sarah A. Lebu, Torsten B. Neilands, Sarah Ssali, Craig R. Cohen, Irene Maeri, Patrick Eyul, Elizabeth A. Bukusi, Edwin D. Charlebois, Carol S. Camlin.

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
