## [Decision Letter · Decision Letter 0]

30 Jun 2022

PGPH-D-22-00458

Sex specific differences in HIV status disclosure and care engagement among people living with HIV in rural communities in Kenya and Uganda

Dear Dr. Okorie,

Thank you for submitting your manuscript to PLOS Global Public Health. After careful consideration, we feel that it has merit but does not fully meet PLOS Global Public Health’s publication criteria as it currently stands. Therefore, we invite you to submit a revised version of the manuscript that addresses the points raised during the review process.

We look forward to receiving your revised manuscript.

Kind regards,

Olatunji O Adetokunboh, MD, PhD

Academic Editor

Journal Requirements:

2. In the online submission form, you indicated that "The datasets generated and analyzed during the current study are available from the corresponding author on reasonable request.". All PLOS journals now require all data underlying the findings described in their manuscript to be freely available to other researchers, either 1. In a public repository, 2. Within the manuscript itself, or 3. Uploaded as supplementary information.

Additional Editor Comments (if provided):

General comments

1. Write out the full meaning of every abbreviation at the first instance including those used in the abstract e.g., HIV.

2. Can you include the justifications for included characteristics of adults who had and had not disclosed HIV-positive status?

3. Blurry figures. Upload better and clearer figures

Reviewers' comments:

Reviewer's Responses to Questions

**Comments to the Author**

1. Does this manuscript meet PLOS Global Public Health’s publication criteria? Is the manuscript technically sound, and do the data support the conclusions? The manuscript must describe methodologically and ethically rigorous research with conclusions that are appropriately drawn based on the data presented.

Reviewer #1: Yes

Reviewer #2: Yes

Reviewer #3: Yes

2. Has the statistical analysis been performed appropriately and rigorously?

Reviewer #1: Yes

Reviewer #2: Yes

Reviewer #3: I don't know

3. Have the authors made all data underlying the findings in their manuscript fully available (please refer to the Data Availability Statement at the start of the manuscript PDF file)?

Reviewer #1: Yes

Reviewer #2: Yes

Reviewer #3: No

4. Is the manuscript presented in an intelligible fashion and written in standard English?

Reviewer #1: Yes

Reviewer #2: Yes

Reviewer #3: Yes

5. Review Comments to the Author

Reviewer #1: Specific Comments

Abstract part

• the abstract past has been developed in somewhat attractive way, but if it includes the aim of the study it will provide some clue to reader about what it is without needing of visiting the research topic.

• It concluded that there is a gender difference on disclosing, care and condom use, but as a recommendation it puts as follow “Interventions focused on differing disclosure support needs for women and men are needed”. So if there is a gender difference, why you have to recommend for the specific gender with a gap?

• The following sentence is duplicated “Unmarried versus married men had higher odds of non48 disclosure (aOR=4.65, 95%CI, 1.32-16.35) and no condom use in the past 6 months 49 (aOR=4.80, 95%CI, 1.74-13.20) and lower odds of receiving HIV care (aOR=0.15; 95%CI, 0.04- 50 0.49).” so revise it.

Methods and material part

• It needs an operational definition for HIV status disclosure

Result part

• Some of the finding are mentioned both in text and in table, so if the response of majority has been mentioned in text then the remain can be described in the table, otherwise it would be a redundancy to the reader. (for instance some the information which are included at table 2 and 3 are already described in text form above the tables)

Discussion part

• I suggest the first paragraph may not be necessary, because it has described at the consecutive paragraph while the specific variable being discussed

Reviewer #2: In the manuscript entitled ‘Sex specific differences in HIV status disclosure and care engagement among people living with HIV in rural communities in Kenya and Uganda’ Okorie et al., have investigated the impact of sex-specific pattern of HIV disclosure among a cohort of PLHIV in Rural Kenya and Uganda. The study documented the sex-specific challenges faced by PLHIV on the issue of HIV disclosure. The important highlight of this study lies in the gender differences in barriers to HIV disclosure, use of condoms and engagement in HIV care bringing to the fore a need for providing different emotional and organizational support for men and women. It is an interesting study conducted with standard methodologies. However, the following comments need to be addressed:

1. Under the introduction section, p.3, line 76, kindly replace “those who choose not to disclose, non-disclosure is a way to” to “those who choose not to disclose, it is a way to”

2. P.3, line 79, please change “study focuses on voluntary disclosure (fully disclosing, selective disclosing, or non-disclosure” to “study focuses on voluntary disclosure (full, selective, or non-disclosure”

3. In the methods section, under data collection, the authors did not mention translating from the local languages used for administering the surveys into English language. I suggest that the authors consider doing so for the sake of clarity.

4. P.15, line 314: please change “When stratified by sex, we see” to “When stratified by sex, we found”

Reviewer #3: In Abstract, may change from ‘Non-disclosure was associated with no past 6-month condom use’ to ‘Non-disclosure was associated with no condom use in past 6 months’ to keep terminology consistent with no condom use and make it less confusing.

Add ‘can’ in following sentence. Otherwise, it infers that non-disclosure always leads to negative outcomes:

Alternatively, non-disclosure, which is often a manifestation of HIV-related stigma (whether internalized, anticipated, or in response to enacted stigma), ‘can’ negatively impacts care outcomes.

Not sure what the following sentence means: ‘Even with high disclosure rates, disclosing is not without its emotional and relational consequences’

Will be good to discuss possible causes of why Non-disclosure is associated with no condom use in the discussion section.

6. PLOS authors have the option to publish the peer review history of their article (what does this mean?). If published, this will include your full peer review and any attached files.

**Do you want your identity to be public for this peer review?** For information about this choice, including consent withdrawal, please see our Privacy Policy.

Reviewer #1: **Yes: **Mebratu Abraha Kebede (Assit. Professor, SPHMMC)

Reviewer #2: No

Reviewer #3: No

---

## [Decision Letter · Decision Letter 1]

16 Jan 2023

PGPH-D-22-00458R1

Sex specific differences in HIV status disclosure and care engagement among people living with HIV in rural communities in Kenya and Uganda

Dear Dr. Okorie,

Thank you for submitting your manuscript to PLOS Global Public Health. After careful consideration, we feel that it has merit but does not fully meet PLOS Global Public Health’s publication criteria as it currently stands. Therefore, we invite you to submit a revised version of the manuscript that addresses the points raised during the review process.

We look forward to receiving your revised manuscript.

Kind regards,

Olatunji O Adetokunboh, MD, PhD

Academic Editor

Journal Requirements:

Additional Editor Comments (if provided):

Dear Authors,

Good revision but need to make the figures clearer.

Reviewers' comments:

Reviewer's Responses to Questions

**Comments to the Author**

1. If the authors have adequately addressed your comments raised in a previous round of review and you feel that this manuscript is now acceptable for publication, you may indicate that here to bypass the “Comments to the Author” section, enter your conflict of interest statement in the “Confidential to Editor” section, and submit your "Accept" recommendation.

Reviewer #4: All comments have been addressed

2. Does this manuscript meet PLOS Global Public Health’s publication criteria? Is the manuscript technically sound, and do the data support the conclusions? The manuscript must describe methodologically and ethically rigorous research with conclusions that are appropriately drawn based on the data presented.

Reviewer #4: Yes

3. Has the statistical analysis been performed appropriately and rigorously?

Reviewer #4: Yes

4. Have the authors made all data underlying the findings in their manuscript fully available (please refer to the Data Availability Statement at the start of the manuscript PDF file)?

Reviewer #4: Yes

5. Is the manuscript presented in an intelligible fashion and written in standard English?

Reviewer #4: Yes

6. Review Comments to the Author

Reviewer #4: (No Response)

7. PLOS authors have the option to publish the peer review history of their article (what does this mean?). If published, this will include your full peer review and any attached files.

**Do you want your identity to be public for this peer review?** For information about this choice, including consent withdrawal, please see our Privacy Policy.

Reviewer #4: **Yes: **Dr Sheriff Abdulrahman

---

## [Editor Report · Decision Letter 2]

3 Mar 2023

Sex specific differences in HIV status disclosure and care engagement among people living with HIV in rural communities in Kenya and Uganda

PGPH-D-22-00458R2

Dear Ms. Okorie,

We are pleased to inform you that your manuscript 'Sex specific differences in HIV status disclosure and care engagement among people living with HIV in rural communities in Kenya and Uganda' has been provisionally accepted for publication in PLOS Global Public Health.

Best regards,

Olatunji O Adetokunboh, MD, PhD

Academic Editor